# 2-Arylpropionic Acid Pyrazolamides as Cannabinoid CB2 Receptor Inverse Agonists Endowed with Anti-Inflammatory Properties

**DOI:** 10.3390/ph15121519

**Published:** 2022-12-06

**Authors:** Daniela R. de Oliveira, Rodolfo C. Maia, Patrícia R. de Carvalho França, Patrícia D. Fernandes, Gisele Barbosa, Lídia M. Lima, Carlos A. Manssour Fraga

**Affiliations:** 1Laboratório de Avaliação e Síntese de Substâncias Bioativas, Instituto de Ciências Biomédicas, Universidade Federal do Rio de Janeiro, Rio de Janeiro 21920-190, RJ, Brazil; 2Programa de Pós-Graduação em Farmacologia e Química Medicinal, Universidade Federal do Rio de Janeiro, Rio de Janeiro 21920-190, RJ, Brazil; 3Laboratório de Farmacologia da Dor e da Inflamação, Instituto de Ciências Biomédicas, Universidade Federal do Rio de Janeiro, Rio de Janeiro 21941-902, RJ, Brazil

**Keywords:** dipyrone, metamizole, cannabinoid receptors, CB_2_ inverse agonists, pyrazolamides, molecular hybridization, NSAID

## Abstract

Among the most recent proposals regarding the mechanism of action of dipyrone, the modulation of cannabinoid receptors CB_1_ and CB_2_ appears to be a promising hypothesis. In this context, the present work describes a series of five novel pyrazolamides (**7**–**11**) designed as molecular hybrids of dipyrone metabolites and NSAIDs, such as ibuprofen and flurbiprofen. Target compounds were obtained in good overall yields (50–80%) by classical amide coupling between 4-aminoantipyrine and arylacetic or arylpropionic acids, followed in some cases by *N*-methylation of the amide group. The compounds presented good physicochemical properties in addition to stability to chemical (pH 2 and 7.4) and enzymatic (plasma esterases) hydrolysis and showed medium to high gastrointestinal and BBB permeabilities in the PAMPA assay. When subjected to functional testing on CB_1_- or CB_2_-transfected cells, compounds demonstrated an inverse agonist profile on CB_2_ receptors and the further characterization of compound LASSBio-2265 (**11**) revealed moderate binding affinity to CB_2_ receptor (K_i_ = 16 µM) with an EC_50_ = 0.36 µM (E_max_ = 63%). LASSBio-2265 (**11**) (at 1, 3, and 10 mg/kg p.o.) was investigated in the formalin test in mice and a remarkable analgesic activity in the late inflammatory phase was observed, suggesting it could be promising for the treatment of pain syndromes associated with chronic inflammatory diseases.

## 1. Introduction

The modulation of the endocannabinoid system has received great focus among therapeutic possibilities for treating pain processes [1]. *In vivo* studies have demonstrated the influence of cannabinoid receptors in mediating nociception [2,3,4,5,6,7,8,9,10]. However, the promotion of central effects attributed to CB_1_ receptor activation, such as hypoactivity, hypothermia, and catalepsy [11,12] which can translate into psychoactivity in humans remains of concern. The use of selective CB_2_ agonists has emerged as alternative, as these compounds attenuate neuropathic pain without the characteristic side effects of CB_1_ activation [13].

The importance of CB_2_ receptors in the modulation of nociception is related to their expression in cells of both the immune and the peripheral nervous system and the relationship of endocannabinoids with decreased inflammation has been demonstrated in several inflammatory pain models [14,15,16,17]. Activation of the endocannabinoid system has also been associated with decreased leukocyte recruitment and increased production of anti-inflammatory cytokines [18].

Dipyrone (**1**) is an analgesic and antipyretic drug with spasmolytic action and is widely used in Brazil, Latin America, Germany, Spain, and some African countries [19]. *In vivo* studies suggest its analgesic effect depends on the activation of cannabinoid receptors [20,21,22], as selective CB_1_ antagonists but not CB_2_ antagonists [23] block the dipyrone-associated anti-hyperalgesia. It is suggested that activation of cannabinoid receptors results from the action of arachidonic acid (**4**)-conjugated dipyrone metabolites, as also proposed for paracetamol [20]. Two major pyrazolamine metabolites are detected after administration of dipyrone to humans: 4-methylaminoantipyrine (**2**), formed after chemical hydrolysis, and 4-aminoantipyrine (**3**), as product of cytochrome P450 (CYP3A4) enzymatic *N*-demethylation (Figure 1) [21]. The biosynthesis of arachidonylpyrazolamide metabolites could result from the action of fatty acid amide hydrolase (FAAH) [22] and these metabolites, henceforward termed (**5**) and (**6**), have demonstrated micromolar affinity for CB_1_ and CB_2_ receptors, although their intrinsic efficacy profiles have not yet been characterized [20].

The present work describes new structural analogs of arachidonylpyrazolamides, (**5**) and (**6**), which exploit the replacement of the arachidonate subunit (A, Figure 2) for a substituted arylacetic or arylpropionic subunit present in some nonsteroidal anti-inflammatory drugs (NSAIDs), especially those belonging to the profen class, due to their well-known relationship with the metabolism of endocannabinoids [24,25].

The use of arylacetic and arylpropionic NSAIDs in the design of the new compounds (**7**–**12**) aimed to mimic the hydrophobic interactions of the alkenyl subunit of arachidonic acid (**4**) with CB_1_ and CB_2_, as previously identified in cyclo-oxygenase (COX) enzymes [26]. Therefore, ibuprofen (**13**) and flurbiprofen (**14**) were selected as representative NSAIDs for the arylpropionic (profen) class and felbinac (**15**), an arylacetic derivative structurally related to (**14**), in order to investigate the influence of the methyl group in the benzylic position. Once synthesized, the target compounds were evaluated regarding their *in vitro* pharmacokinetic profiles and then characterized for their *in vitro* intrinsic efficacy. Moreover, the best compound was submitted to a classical *in vivo* model to confirm its potential as an analgesic drug candidate that acts as a modulator of cannabinoid receptors for the treatment of acute and chronic pain states.

## 2. Results and Discussion

### 2.1. Chemical Synthesis of NSAID-Pyrazolamides (**7**–**12**)

The target unsubstituted pyrazolamide derivatives (**7**–**9**) were synthesized through the classical coupling of a carboxylic acid group of the NSAIDs (**13**–**15**) with 4-aminoantipyrine (**3**) using *N*,*N*′-carbonyldiimidazole (CDI) and 1-hydroxybenzotriazole (HOBT) as coupling reagents in the presence of triethylamine [27], as illustrated in Figure 1. The desired pyrazolamides (**7**–**9**) were obtained in yields ranging 52–66%.

Next, pyrazolamides (**7**–**9**) were submitted to an *N*-methylation step after treatment with sodium hydride in THF [28], followed by the addition of methyl iodide, as illustrated in Figure 2. By using these conditions, pyrazolamides **10** and **11** derived from ibuprofen and flurbiprofen, respectively, were prepared in 75% and 79% yields. On the other hand, the expected *N*-methyl carboxamide (**12**) derived from **9** could not be obtained using this methodology, and instead, the bismethylated derivative (**16**) was formed in a 56% yield (Figure 2). This behavior was associated to the similar pKa values between the amide and the α-carbonyl hydrogens of **9** [29,30], favoring deprotonation on both sites and bisalkylation to provide compound **16**. Pyrazolamides **7** and **8** were not susceptible to alkylation at the benzylic position due to the steric effect produced by the methyl group originally present.

Attempts to circumvent this problem by changing the base to potassium carbonate in acetone were not successful, and the only product obtained after 48 h was pyrazolamide **16** in low yields. In addition, the use of another strategy that foresaw the initial methylation of Boc-protected **3**, followed by deprotection and coupling with **15**, was also unsuccessful due to the low reactivity of *N*-Boc-pyrazolamide when subjected to alkylation (see Appendix A). Therefore, considering that the main motivation for the synthesis of **12** was to evaluate how the absence of the methyl group in the benzylic position could influence the modulation of cannabinoid receptors, we decided to continue the investigation with only the five initially planned pyrazolamides (**7**–**11**).

Compounds **7**–**11** were fully characterized using spectroscopic techniques (see Appendix A), and their purity was determined by reverse-phase HPLC analysis and high-resolution mass spectrometry to be greater than 95%, which was considered adequate for the next step of investigating their drug-like properties and biological actions on cannabinoid receptors.

### 2.2. Drug-like Properties of Pyrazolamides **7**–**11**

After full characterization of compounds **7**–**11**, we proceeded to determine their physicochemical and *in vitro* pharmacokinetic properties, given their strategic relevance for developing novel orally available drug candidates.

Aqueous solubility of pyrazolamides **7**–**11** was determined at pH 7.4 in phosphate buffer, yielding values within 10–100 µM (Table 1) [31]. These values are considered a reasonable range for drug candidates, since either low (<10 μM) or elevated aqueous solubility (>100 µM) can result in limited intestinal absorption [32]. Octanol/phosphate buffer (pH 7.4) partition coefficients (Log D_7.4_) of **7**–**11** remained are within the ideal range for drug candidates, i.e., 2–3 [33,34].

The membrane permeability of **7**–**11** measured using a gastrointestinal tract (GI) parallel artificial membrane permeability assay [35] (PAMPA) ranged from 2.5–15.3 ×10^−6^ cm/s, which indicated excellent absorption rates (>60%) after oral administration of these substances (Table 1). In addition, considering that cannabinoid receptors are also expressed in the central nervous system [36], the PAMPA-BBB assay was also performed to determine the potential of these compounds to cross the blood–brain barrier [37,38] (Table 1). As expected, all the pyrazolamides had a permeability rate compatible with accessing the CNS (CNS+).

Although the results suggested elevated absorption of **7**–**11**, clearance rates are also of concern for ideal bioavailability. Therefore, we also investigated the stability of **7**–**11** against enzymatic hydrolysis, as amide groups are potentially labile to plasma carboxylesterases [39] (Table 1). Flurbiprofen-derived pyrazolamides (**8** and **11**) showed increased stability against enzyme hydrolysis than corresponding ibuprofen-derived pyrazolamides (**7** and **10**), possibly by the steric effect of the bulkier and less flexible biphenyl system compared to 4-isobutylphenyl in the latter. The loss of ring coplanarity also contributes to the greater plasma stability of both **8** and **11**, as the absence of the *ortho*-effect in **9** favors its metabolism, as shown in Table 1.

Furthermore, considering that the compounds developed in the present work have amide functional groups, the stabilities of the compounds were evaluated at gastric (pH 2) (Figure 3A) and plasmatic pH (7.4) (Figure 3B) to assess their susceptibility to pH-dependent chemical hydrolysis. After 240 min, derivatives **7**–**11** showed recovery rates of 70-100%, indicating the high chemical stability of these compounds at both pH values [40]. As expected, tertiary pyrazolamides **10** and **11** proved to be more resistant to hydrolysis than the corresponding secondary amides.

Therefore, both absorption rates and chemical and plasma stability of **7**–**11** supported the possibility of their use by oral route, with flurbiprofen-derived pyrazolamides **8** and **11** presenting the best physicochemical (solubility, lipophilicity) and *in vitro* pharmacokinetic profile (permeability and stability) in comparison to the other analogs evaluated.

### 2.3. In Vitro Activity Profiles of Pyrazolamides **7**–**11**

The agonist profile of pyrazolamides **7**–**11** towards cannabinoid receptors was investigated by measurement of cyclic AMP (cAMP) concentrations in CHO cells transfected with either human CB_1_ or CB_2_. The assay is based on receptor coupling to G proteins of the Gi/o family, whose activation results in adenylate cyclase inhibition and a decrease in intracellular cAMP [41,42,43]. When evaluated in CB_1_-expressing cells, compounds **7**–**11** did not significantly altered the intracellular cAMP at either 1 µM or 10 µM [44], as the reduction in cAMP concentration did not exceed 25% of the maximum response obtained with control agonist CP-55,940 (Figure 4 and Appendix A). Results indicate a concentration-dependent weak response and, although no structure-activity relationship was evidenced, these findings demonstrate that introduction of the methyl group in **10** and **11** did not result in a significant change in the activity profile of these compounds against CB_1_ receptors compared to **7** and **8**, respectively.

On another hand, when evaluated in cells expressing human CB_2_ receptors, compounds **7**–**11** showed a significant increase in cAMP concentrations when tested at 10 µM [44] (Figure 5). This result was in marked contrast with the response to 100 nM WIN 55212-2, which completely abolished the production of cAMP by forskolin, indicating a concentration-dependent inverse agonist profile for **7**–**11**. Additionally, LASSBio-2265 (**11**) showed an important effect starting from 1 μM, while no effect was observed in CB_1_-expressing cells.

Considering the ensemble of results, LASSBio-2265 (**11**) was selected for further investigation to confirm the hypothesis of CB_2_ inverse agonism. A concentration-response curve for the cAMP increase induced by compound **11** was assessed using chemiluminescent detection in CHO-K1 CNR2 Gi cells stimulated with 8 μM (EC_20_) forskolin, validated against full agonist CP 55,940 and inverse agonist SR 144,528 [45]. The functional status of the CB_2_ receptor was monitored by cAMP quantification in relation to controls and it was possible to observe an increase in cAMP by DiscoverX HitHunter cAMP XS+ assay. The obtained data confirmed that LASSBio-2265 (**11**) is a partial inverse agonist of human CB_2_ receptors, with an EC_50_ of 0.369 ± 0.03 μM and an efficacy of 63% of the maximum effect.

Moreover, the binding affinity of LASSBio-2265 (**11**) to human CB_2_ receptors was also studied to further confirm if the inverse agonist activity of LASSBio-2265 (**11**) was indeed product of an interaction with CB_2_ receptors [46]. Competitive binding was evaluated in membranes from CB_2_-expressing CHO cells using [^3^H]-WIN55212-2 as radioligand probe. The results indicated that **11** shows a moderate affinity for CB_2_ receptors, with Ki = 16.0 ± 3 µM. However, the affinity obtained for compound **11** is of similar magnitude to those found by Rogosch and coworkers for compounds **5** and **6**, as illustrated in Table 2. Potency and affinity values were determined from *in vitro* assays performed in duplicate.

### 2.4. In Vivo Activity of LASSBio-2265 (**11**) in the Formalin-Induced Licking Response in Mice

Selective inverse agonists of CB_2_ receptors, JTE-907 [47,48], SMM-189 [49], and Sch.414319 [50], have been indicated in the literature as beneficial agents for the treatment of diverse pathological conditions associated with inflammatory processes. Cascio and co-workers [51] described the antinociceptive activity of CB_2_ receptor inverse agonists in the late phase of the classical formalin test, which induces a biphasic stereotypical nocifensive behaviour. Therefore, the analgesic profile of LASSBio-2265 (**11**) was evaluated in the same model (Figure 6), which measures nociceptive behavior after subcutaneous (s.c.) injection of dilute formalin (1.25% in saline, 30 μL) into one of the two hind paws of mice [52]. This assay presents two temporally distinct phases. The first (early) phase evokes nociception through the direct effect of formalin on the acute activation of pain-sensing C fibers at the peripheral endings of sensory neurons involved in pain transmission. After an interval of 10–15 min, a second, inflammatory-driven phase (late phase) of sustained pain behavior appears, in which sensory fiber activity is accompanied by inflammation and central sensitization [53].

In the first phase, only morphine (2.5 mg/kg) promoted significant analgesia compared to vehicle (5% DMSO in 0.9% NaCl), acetylsalicylic acid (ASA, 200 mg/kg) and LASSBio-2265 (**11**) at 1, 3 and 10 mg/kg. In contrast, during the second phase, LASSBio-2265 (**11**) proved to be comparable to both morphine and ASA, with dose-dependent progressive effect, thus proving to be promising for the treatment of conditions that lead to inflammatory pain. Naïve animals were evaluated in two phases of assay without any treatment (Figure 6).

## 3. Materials and Methods

### 3.1. Materials

Commercial reagents were obtained from Sigma–Aldrich (St. Louis, MI, USA). Melting points were determined by differential scanning calorimetry with a Shimadzu calorimeter (Model DSC-60). Fourier-transformed infrared (FTIR) spectra were obtained with ThermoScientific spectrometer (Model Nicolet iS10). ^1^H- and ^13^C-NMR spectra were obtained with VARIAN 400-MR and 500-MR spectrometers. High-performance liquid chromatography with detection by photodiode arrays (HPLC-PDA) was performed with Shimadzu LC20AD apparatus using a Kromasil 100-5C18 column (4.6 mm × 250 mm) and SPD-M20A detector (Diode Variety). Ultraviolet-visible measurements were performed in a Femto scanning spectrophotometer (Model 800XI). High-resolution mass spectrometry (Orbitrap-HRMS) analysis was performed using a QExactive Hybrid Quadrupole Orbitrap Mass Spectrometer (Thermo Fisher Scientific, Waltham, MA, USA) with electrospray ionization (ESI) using solutions of the compounds (1 µg/mL) prepared in a 7:3 ratio of water:methanol fortified with 0.1% formic acid and 5 mM ammonium formate.

### 3.2. Synthesis of the NSAID-Pyrazolamides (**7**–**11**)

#### 3.2.1. Synthesis of the NSAID-Pyrazolamides (**7**–**9**)

To a mixture of 0.95 mmol of carboxylic acid (13), (14), or (15), 185 mg (1.14 mmol, 1.2 eq.) of 1,1^′^-carbonyldiimidazole, 154 mg (1.14 mmol, 1.2 eq.) of hydroxybenzotriazole, and 260 μL (1.88 mmol, 2 eq.) of triethylamine, 290 mg (1.42 mmol, 1.5 eq.) of 4-amino-antipyrine (**3**) dissolved in 15 mL of dichloromethane was added. The reaction was magnetically stirred for 4 h at room temperature. After this time, the reaction mixture was poured into a separatory funnel and extracted with dichloromethane:water (2:3). The organic phase was treated with anhydrous sodium sulfate and concentrated under reduced pressure. The desired pyrazolamide (**7**–**9**) was obtained after purification by isocratic elution with ethyl acetate:hexane (9:1, *v/v*) from a flash silica gel column (1077341003; Millipore, Burlington, VT, USA), using aprox. 1 g of silica per mg of raw product.

*N*-(1,5-dimethyl-3-oxo-2-phenyl-2,3-dihydro-1*H*-pyrazol-4-yl)-2-(4-isobutylphenyl) propanamide (**7**). White powder, 66% yield, m.p.: 151.2 °C. ^1^H-NMR (400 MHz, DMSO-*d*_6_) δ (ppm): 9.18 (s, 1H); 7.5 (d, *J* = 7.0 Hz 2H); 7.4 (m, 2H); 7.31 (d, *J* = 7.3 Hz 2H), 7.10 (m, 3H); 7.08 (d, *J* = 7.0 Hz 2H), (s, 3 H); 3.82 (q, *J* = 7.0 Hz 1H); 3.00 (s, 1H); 2.41 (d, *J* = 7.1 Hz 2H); 1.97 (s, 3H); 1.81 (m, *J* = 6.9 Hz 1H); 1.36 (s, 1H); 0.86 (s, 3H); 0.84 (s, 3H). ^13^C-NMR (100 MHz, DMSO-*d*_6_) δ (ppm): 173.08 (C10); 161.80 (C15); 152.22 (C5); 139.36 (C2); 139.32 (C18); 135.05 (C21); 129.07 (C23, C25); 128.82 (C1, C3); 126.98 (C24); 126.16 (C4, C6); 123.39 (C22, C26); 44.39 (C14); 44.26 (C7); 36.06 44.39 (C9); 29.65 (C20); 22.20 (C8); 22.18 (C28, C29); 18.70 (C11); and 11.08 (C19). IR (ATR, cm^−1^): 1682.06 (υ C=O); 3224.63 (υ N-H); and 2961.65 (υ C-H). ESI-FT-ICR, [M + H]^+^ (calculated): 392.23325 *m/z*, [M + H]^+^ (experimental): 392.23268 *m/z*.

*N*-(1,5-dimethyl-3-oxo-2-phenyl-2,3-dihydro-1*H*-pyrazol-4-yl)-2-(2-fluoro-[1,1^′^-biphenyl]-4-yl il) propanamide (**8**). White powder, 66% yield, m.p.: 214.8 °C. ^1^H-NMR (400 MHz, DMSO-*d*_6_) δ (ppm): 9.35 (s, 1H); 7.54 (d, *J* = 8.6 Hz, 2H); 7.46 (m, 2H); 7.48 (d, *J* = 7.1 Hz, 2H); 7.36 (m, 1H); 7.4 (m, 1H); 7.31 (m, 3H); 3.93 (q, *J* = 6.8 Hz, 1H); 3.00 (s, 3H); 2.04 (s, 3H); and 1.43 (d, *J* = 6.9 Hz, 3H). ^13^C-NMR (100 MHz, DMSO-*d*_6_) δ (ppm): 172.37 (C14); 161.71 (C19); 157.85 (C3); 152.01 (C5); 135.01 (C7); 130.55 (C22); 130.52 (C22′); 129.06 (C25); 128.71 (C28); 128.69 (C28′); 128.60 (C27, C29); 127.74 (C9, C11); 126.20 (C10); 123.85 (C1); 123.44 (C8,C12); 114.98 (C26, C30); 114.79 (C18); 107.41 (C6); 44.24 (C13); 36.03 (C24); 18.54 (C15) and 11.22 (C23). IR (ATR, cm^−1^): 1680.78 (υ C=O amide); and 3196.11 (υ N-H). ESI-FT-ICR, [M + H]^+^ (calculated): 430.19253 *m/z*, [M + H]^+^ (experimental): 430.19174 *m/z*.

2-([1,1’-biphenyl]-4-yl)-*N*-(1,5-dimethyl-3-oxo-2-phenyl-2,3-dihydro-1*H*-pyrazol-4-yl) acetamide (**9**). Yellow powder, 56% yield, m.p.: 221.8 °C. ^1^H-NMR (400 MHz, DMSO-*d*_6_) δ (ppm): 9.35 (s, 1H); 7.66 (m, 2H); 7.62 (m, 2H); 7.51 (d, *J* = 7.7 Hz, 2H); 7.47 (s, 1H); 7.42 (m, 2H); 7.35 (t, *J* = 7.4 Hz, 1H); 7.3 (m, 3H); 3.65 (s, 2H); 3.03 (s, 3H); and 2.07 (s, 3H). ^13^C-NMR (100 MHz, DMSO-*d*_6_) δ (ppm): 169.73 (C14); 161.81 (C18’); 152.25 (C2); 129.39 (C5); 129.38 (C21); 129.34 (C24); 129.33 (C26, C28); 129.19 (C9, C11); 128.74 (C27); 128.73 (C4, C6); 126.77 (C10); 126.38 (C8, C12); 107.52 (C1, C3); 41.68 (C25, C29); 35.78 (C17); 11.33 (C13) and 11.17 (C23). IR (ATR, cm^−1^): 1680.29 (υ C=O) and 3304.28 (υ N-H). ESI-FT-ICR, [M + H]^+^ (calculated): 398.18630 *m/z*, [M + H]^+^ (experimental): 398.18582 *m/z*.

#### 3.2.2. Synthesis of N-Methylated Pyrazolamides (**10**) and (**11**)

To a round-bottomed balloon containing 0.25 mmol of compound (**7**) or (**8**), 1.5 eq. of sodium hydride (from a 60% dispersion in mineral oil dissolved in 5 mL of anhydrous tetrahydrofuran) was added under an argon atmosphere. After 15 min at 0 °C, 47 μL (0.75 mmol, 3 eq.) iodomethane was added, and the reaction was monitored by TLC until completion (aprox. 4 h). The reaction mixture was poured into a separatory funnel and extracted with dichloromethane:water (2:3). The organic phase was treated with anhydrous sodium sulfate and concentrated under reduced pressure. The desired pyrazolamide (**10** or **11**) was obtained after purification by isocratic elution with ethyl acetate:hexane (9:1, *v/v*) from a flash silica gel column (1077341003; Millipore, USA), using aprox. 1 g of silica per mg of raw product [28].

*N*-(1,5-dimethyl-3-oxo-2-phenyl-2,3-dihydro-1*H*-pyrazol-4-yl)-2-(4-isobutylphenyl)-*N*-methylpropanamide (**10**). Yellow oil. ^1^H-NMR (400 MHz, DMSO-*d*_6_) δ (ppm): 7.57 (s, 1H); 7.55 (s, 1H); 7.53 (d, *J* = 8.6 Hz, 2H); 7.39 (d, *J* = 7.3 Hz, 2H); 7.09 (s, 1H); 7.07 (s, 1H); 6.89 (d, *J* = 7.0 Hz 1H); 3.72 (q, *J* = 6.7 Hz, 1H, H2); 2.97 (s, 3H); 2.95 (s, 3H); 2.40 (d, *J* = 7.0 Hz, 2H); 1.80 (m, 1H); 1.33 (s, 3H); 1.20 (d, *J* = 6.8 Hz, 3H); and 0.87 and 0.85 (s, 3H). ^13^C-NMR (100 MHz, DMSO-*d*_6_) δ (ppm): 174.36 (C10); 161.39 (C15); 153.53 (C5); 139.73 (C2); 134.76 (C18); 129.21 (C21); 129.12 (C1, C3); 129.00 (C23, C25); 126.80 (C24); 124.19 (C14); 112.61 (C22, C26); 44.26 (C7); 42.92 (C9); 35.94 (C20’); 35.02 (C30); 29.63 (C8); 22.10 (C28, C29); 20.51 (C11); and 9.13 (C19). IR (ATR, cm^−1^): 1655.36 (υ C=O) and 2925.59 and 2956.95 (υ C-H). ESI-FT-ICR, [M + H]^+^ (calculated): 406.24890 *m/z*, [M + H]^+^ (experimental): 406.24832 *m/z*.

*N*-(1,5-dimethyl-3-oxo-2-phenyl-2,3-dihydro-1*H*-pyrazol-4-yl)-2-(2-fluoro-[1,1’-biphenyl]-4-yl)-*N*-methylpropanamide (LASSBio-2265) (**11**). Yellow oil. ^1^H-NMR (500 MHz, CDCl_3_) δ (ppm): signals unfolded due to the presence of 2 rotamers (approximate 4:1 ratio) 7.5 (m, 2H); 7.37 (m, 3H); 7.34 (m, 2H); 6.98 (m, 1H); 6.90 (m, 3H); 6.85 (m, 3H) 3.80 (q, *J* = 7.0 Hz, 1H, minor); 3.72 (q, *J* = 6.9 Hz, 1H, major); 3.14 (s, 3H); 3.00 (s, 1H); 1.45 (s, 3H, major); 1.42 (s, 3H, minor); 1.26 (d, 1H); and 1.23 (d, 1H). ^13^C-NMR (100 MHz, acetonitrile-*d*_3_) δ (ppm): 175.50 (C14); 163.01 (C3); 161.67 (C19); 159.24 (C5); 154.57 (C22); 131.99 (C7); 130.25 (25); 129.94 (C28); 129.91 (C28′); 129.59 (C27, C29); 128.84 (C9, C11); 128.13 (C10); 125.65 (C18); 124.75 (C8, C12); 115.82 (C1); 115.59 (C26, C30); 113.83 (C6); 44.38 (C13); 36.81 (C24); 35.90 (C33); 20.58 (C15) and 10.15 (C23). IR (ATR, cm^−1^): 1665.03 (υ C=O) and 2980.12 and 2932.32 (υ C-H). ESI-FT-ICR, [M + H]^+^ (calculated): 444.20818 *m/z*, [M + H]^+^ (experimental): 444.20806 *m/z*.

Structural characterization of pyrazolamide (**7**–**11**) is shown in Appendix A.

### 3.3. Drug-like Properties of NSAID-Pyrazolamides (**7**–**11**)

#### 3.3.1. Aqueous Solubility

The test was performed based on the absorbance obtained by ultraviolet spectroscopy. The scanning wavelength was determined according to the highest absorbance observed for each compound. Saturated aqueous solution of the analyzed compounds were stirred for 4 h at 37 °C, and then filtered through a 0.45 µm PVDF filter and transferred to a quartz cuvette (10 mm optical path) for reading. Solubility was determined by linear regression based on the equation of the line [31].

#### 3.3.2. Determination of the Distribution Coefficient at pH 7.4 (log D_7.4_) 

The assay was performed based on the shake-flask method and the absorbance readings by ultraviolet spectroscopy at maximum absorbance wavelength. To a Falcon tube with a capacity of 15 mL, 5 mL of buffer solution at pH 7.4 and 5 mL of 1-octanol were added. Then, the tested compounds were added at defined concentrations of approximately 10–15 µM, and the mixtures were homogenized via vigorous manual shaking. Next, the Falcon tubes were stirred for 4 h at 37 °C, and the experiment was carried out in triplicate for each compound. Subsequently, each sample was placed in a separatory funnel, and the collected aqueous phase was filtered through a 0.45 µm PVDF filter and transferred to a quartz cuvette (10 mm optical path) for UV/Vis reading. The distribution coefficient of each compound at pH 7.4 was determined by linear regression using the straight-line equation [33].

#### 3.3.3. Chemical Stability 

To a 2 mL microtube, 2 μL of a 5 mM concentrated solution of the analyzed compound (solubilized in DMSO) and 248 μL of acid buffer (0.2 M potassium chloride and 0.2 M HCl; pH = 2) or neutral (dibasic phosphate; pH = 7.4) were added. The experiment was carried out in triplicate. After vortexing, the mixture was placed in a water bath at 37 °C under vigorous stirring for 0, 30, 60, 120, and 240 min. After each reaction time, 248 μL of basic buffer (phosphate buffer, pH = 7.4) was added to neutralize the pH of the medium in the experiments using acidic buffer (pH = 2). Extraction of the compound was carried out by adding 1 mL of acetonitrile. The entire contents of the microtube were vortexed vigorously, and the container was placed in a freezer (−20 °C) to freeze the aqueous phase. Then, the organic phase was separated, filtered through an HPLC filter with a 0.45 μm PVDF membrane of 25 mm in diameter, and analyzed by HPLC-PDA (mobile phase: acetonitrile:water 60:40). The injection volume of the samples was 20 μL at a flow rate of 1.0 mL/min. Quantitation was performed at 272 nm for compounds (**7**), (**8**), and (**10**), 247 nm for compound (**8**) and LASSBio-2265 (**11**), and 253 nm for compound (**9**) [40].

#### 3.3.4. Parallel Artificial Membrane Permeability Assay (PAMPA-GTI) 

A 10 mM solution of each compound (tests and controls) was prepared in DMSO. Then, in a 5 mL glass flask, 250 µL of the freshly prepared solution was homogenized with 4750 µL of PBS at pH 6.6 to a concentration of 10 mM. The solution was then filtered in a 0.45 µm PVDF filter and set aside. Then, 180 µL of a solution of phosphate-bufffered saline (PBS, pH 7.4):DMSO (95:5) was added to the wells of the receptor plate, and 5 µL of the soy L-α-phosphatidylcholine lipid solution (20 mg/mL in dodecane) was added to the wells of the donor plate. After 5 min, the donor plate received 180 µL of the reserved solution containing each compound. Afterward, the donor plate was carefully placed on top of the recipient, and the plates were agitated at 50 rpm for 8 h at room temperature (±25 °C) in a closed container containing 10 mL of PBS at pH 7.4. After this period, the donor plate was removed, and the contents of the recipient plate were transferred to a UV reading plate and read (SpectraMax 5, Molecular Devices) at the maximum wavelengths for each compound. The blank was prepared with 180 µL of PBS solution (pH 7.4):DMSO (95:5). The experiments were carried out in triplicate with two different analyses (n = 2) [37]. The optical density values obtained from the readings at each selected wavelength for each of the compounds were analyzed by comparison with the values of the controls. These values were used to determine the absorbed fraction (Fa %) using the JMP^®^ program version 10.0. The permeability results from PAMPA-TGI [35,54] classified the compounds according to the absorbed fraction percentage (Fa%) as follows: high intestinal permeability: >70%; medium intestinal permeability: 30–69%; or low intestinal permeability: <29% [55].

#### 3.3.5. Parallel Artificial Membrane Permeability Assay (PAMPA-BBB) 

In a 5 mL glass vial, 1 mg of each compound (test or control) was dissolved in 1 mL of ethanol. Then, 500 μL of ethanol and 3.5 mL of PBS at pH 7.4 were added to this solution. The solution was then filtered in a 0.45 µm PVDF filter and set aside. Subsequently, 180 μL of a solution of PBS (pH 7.4): ethanol (70:30) was added to the wells of the receptor plate, and 5 μL of porcine brain lipid solution (20 mg/mL in dodecane) was added to the wells of the donor plate. After 5 min, the donor plate received 180 μL of each compound solution in triplicate. Then, the donor plate was carefully placed on top of the recipient plate, forming a sandwich system, and the plates were left to rest for 2 h and 45 min at room temperature (±25 °C) in a closed container containing 10 mL of PBS at pH 7.4. After this period, the donor plate was removed, and the contents of the recipient plate were transferred to a UV reading plate and read (SpectraMax 5, Molecular Devices) at the maximum wavelengths for each compound. The blank was prepared with 180 µL of PBS solution (pH 7.4):ethanol (70:30) [56]. The experiments were performed in triplicate with two different analyses (n = 2). Assay validation was performed using atenolol, caffeine, diazepam, enoxacin, ofloxacin, testosterone, and verapamil, which are reference compounds with described permeability values and were used as standards. Results indicated (a) CNS+ (high predicted BBB permeability) Pe > 4.0 × 10^−6^ cm/s; (b) CNS− (low predicted BBB permeability) Pe < 2.0 × 10^−6^ cm/s; and (c) CNS+/− (uncertain permeability) Pe between 2.0–4.0 × 10^−6^ cm/s [56,57].

#### 3.3.6. Plasmatic Stability 

For the evaluation of plasma stability against plasma carboxylesterases, a pool of plasma male Wistar rats weighing 300–360 g kept in the vivarium of the Laboratory of Biochemical and Molecular Pharmacology-UFRJ was used. Blood was collected with heparin and kept on ice. Then, the samples were centrifuged at 2000 rpm for 15 min at 20 °C, with subsequent removal of the plasma. The plasma was then diluted 80% (*v*/*v*) with phosphate buffer (PBS; pH = 7.4) at 37 °C. The experiments performed in this work were approved by the Ethics Committee on the Use of Animals in Scientific Experiments of the Health Sciences Center of the Federal University of Rio de Janeiro (Number 028/15). The biological matrix was incubated at 37 °C under agitation for 30, 60, 120, and 240 min. The degree of degradation was evaluated by HPLC-PDA with methyl 1,1′bisphenyl-4-carboxylate as an internal standard (PI), using an isocratic mobile phase of 70%:30% acetonitrile:water (0–15 min), an injection volume of 20 μL, a flow rate of 1 mL/min, and detection at 254 nm [39].

### 3.4. In Vitro Experiments

#### 3.4.1. Agonist Activity toward CB1 and CB2 Receptors

The cellular functional assay used to evaluate the agonist activity of planned pyrazolamides (**7**–**11**) toward cannabinoid receptor subtypes 1 and 2 was carried out at Eurofins Discovery France (https://www.eurofinsdiscoveryservices.com/) under the study number 100050410 (accessed on 28 October 2019) using the protocol previously published by Felder and coworkers [44]. Briefly, pyrazolamides **7**–**11** were assayed at concentrations of 1 µM and 10 µM to evaluate their ability to change intracellular cAMP in CHO cells transfected with human CB_1_ and CB_2_ receptors [44], for 30 or 10 min at 37 °C after stimulation with 10 μM forskolin, respectively. Negative and blank controls were performed in triplicate with and without the addition of forskolin, respectively. Positive controls used were CB agonists CP-55,940 (at 1 nM) and WIN 55,212-2 (at 100 nM).

#### 3.4.2. Inverse Agonist Activity toward CB2 Receptors [45]

The cellular functional assay used to evaluate the inverse agonist activity of LASSBio-2265 (**11**) toward the cannabinoid CB_2_ receptor was carried out at Eurofins DiscoveryX Corporation USA (https://www.discoverx.com/home) under the study number FR095-0025341-Q (accessed on 02 June 2021) using the GPCR Biosensor Assay. Cell line cAMP Hunter CHO-K1 CNR2 Gi was seeded at a total volume of 20 µL into white-walled, 384-well microplates and incubated at 37 °C prior to testing in the presence of 8 μM (EC_20_) forskolin. The media was aspirated from the cells and replaced with 15 µL a 2:1 solution of Hanks’ balanced salt solution (HBSS)/10 mM Hepes:cAMP XS+ Ab reagent. Intermediate dilution of the sample stocks was performed to generate 4× samples in assay buffer containing 32 μM forskolin. Then, 5 µL of each 4× sample was added to the cells and incubated at 37 °C for 60 min. The final assay vehicle concentration was 1%. After incubation, the assay signal was generated by incubation with 20 µL cAMP XS+ ED/CL lysis cocktail for 1 h followed by incubation with 20 µL cAMP XS+ EA reagent for 3 h at room temperature. After signal generation, the microplates were read with a PerkinElmer Envision instrument for chemiluminescent signal detection. Compound activity was analyzed using the CBIS data analysis suite (ChemInnovation, San Diego, CA, USA). Percent activity was calculated using the following formula:

% Inverse Agonist Activity = 100% × ((mean RLU of test sample − mean RLU of EC_20_ forskolin)/(mean RLU of forskolin positive control − mean RLU of EC_20_ control)).

Data obtained from duplicate experiments were fit to concentration-response curves by non-linear regression of the Hill equation. Curve-fit parameters and standard errors were calculated using GraphPad Prism 6.0 (Hercules, CA, USA).

#### 3.4.3. Human CB2 (Agonist Radioligand) Receptor Binding Assay

Assays of binding to CB_2_ receptors were performed at Eurofins Discovery France (https://www.eurofinsdiscoveryservices.com/) under the study number 100057099 (accessed on 3 June 2021) using the protocol previously published by Munro, Thomas, and Abu-Shaar [46]. Human recombinant CB_2_ receptors were expressed in CHO cells, and binding was performed with [^3^H]-WIN 55212-2, which is a nonselective agonist radioligand. The analysis was performed using software developed at Cerep (Hill software, Santa Barbara, CA, USA) and SigmaPlot 4.0 for Windows (^©^1997 by SPSS Inc., Chicago, IL, USA).

Data obtained from duplicate experiments were fit to concentration-response curves by non-linear regression of the Hill equation. Curve-fit parameters and standard errors were calculated using GraphPad Prism 6.0 (Hercules, CA, USA).

### 3.5. In Vivo Experiments

#### 3.5.1. Animals

Swiss Webster mice (20–25 g, 30–40 days old), from both sexes were donated by the Instituto Vital Brazil (Niteroi, Rio de Janeiro, Brazil). The mice were maintained in a room with a 12 h light-dark cycle at 22 ± 2 °C and 60% to 80% humidity, with food and water provided *ad libitum*. The mice were acclimatized to the laboratory conditions for at least 1 h before each test onset and were used only once throughout the experiments. All protocols were conducted in accordance with the Guidelines on Ethical Standards for Investigation of Experimental Pain in Mice (Zimmermann, 1983) and followed the principles and guidelines adopted by the National Council for the Control of Animal Experimentation (CONCEA) approved by the Ethical Committee for Animal Research (CEUA, approval in 30 April 2019, receiving the number 31/19). All experimental protocols were performed during the light phase. The animal numbers per group were kept to a minimum, and at the end of each experiment, the mice were euthanized by ketamine/xylazine overdose.

#### 3.5.2. Formalin-Induced Licking Behavior

This assay was performed as described by Hunskaar et al. [53] and adapted by Gomes et al. [58]. This model was characterized by a response that occurred in two phases. The first phase (acute neurogenic pain) occurred during the first 5 min after the intraplantar injection of formalin, and the second phase (inflammatory pain) occurred 15 to 30 min post-injection. Mice were pretreated with oral doses of LASSBio-2265 (**11**, at 1, 3 or 10 mg/kg, p.o.), morphine (2.5 mg/kg, i.p.), acetylsalicylic acid (ASA, 200 mg/kg, p.o.), or the vehicle (same amount of DMSO used to solubilize the substances + NaCl 0.9%) 60 min before the administration of formalin. The naïve group was composed of animals that did not receive any oral treatment. Then, mice (n = 7 per group) received 20 μL of formalin (2.5% *v/v*) in the dorsal surface of the left hind paw. The time that the mice spent licking the injected paw was immediately recorded. The amount of DMSO used did not affect or interfere per se with the experimental model. The final volume administered to mice was 0.1 mL. The protocol was carried out by a blinded experimenter that did not know the drug treatment conditions and groups.

## 4. Conclusions

NSAID-pyrazolamides **7**–**11** were obtained in good yields from classical synthetic methodologies. All compounds presented adequate aqueous solubility, lipophilicity, permeability, and stability to chemical and enzymatic hydrolysis for the development of orally administered drug candidates. Compounds **7**–**11** also showed high permeability through artificial membranes mimicking the blood–brain barrier, as cannabinoid receptors are widely distributed in the central nervous system. Their stability against carboxylesterases also reinforces that observed effects are not resulting from the effect of precursors, i.e., 4-aminoantipyrine and NSAID. The chemical stability of pyrazolamides **7**–**11** indicated that the compounds are highly stable and are not susceptible to pH-dependent hydrolysis, unlike dipyrone.

The *in vitro* experiments performed for characterization of the intrinsic efficacy of LASSBio-2265 (**11**) in CB_2_ receptors showed a moderate ligand profile (Ki = 16 µM) with EC_50_ = 0.36 µM (Emax = 63%) in CHO cells. For *in vivo* studies, antinociceptive effects of **11** were demonstrated by decreased pain behaviour in the formalin test in mice at 3 mg/kg, suggesting that it could be a promising candidate for the treatment of pain syndromes associated with chronic inflammatory diseases.

## Data Availability

Not applicable.

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
