# Peer review of "2-Arylpropionic Acid Pyrazolamides as Cannabinoid CB2 Receptor Inverse Agonists Endowed with Anti-Inflammatory Properties"

_pharmaceuticals, 2022, doi:10.3390/ph15121519_

Round 1
Reviewer 1 Report
The paper describes molecular hybrids of dypirone and NSAID as potential pharmaceutical candidates for pain management. The research is interesting. However there are series of methodological questions to be addressed by the authors.
Major issues:
2.3. Section of the paper should be rephrased to have more insight about the methodical procedure. I believe that the pharmaceutical compounds are given after forskolin treatment, but its not elaborated in the manuscript. Neither the number of measurements is given nor the SD/SEM and the results of the statistical analysis is indicated in Fig 4 and 5. The figures should be at least partially self explanatory, but this is not the case here.
There is no detailed description of the results of LASSBio-2265 as a CB2 inverse agonist. (eg. how many measurements are done).
The same problem is seen in the radioligand experiments as well.
2.4. Please describe the methodology in a more detailed way. Were the behavior measurements in the 2nd phase were done between 15-30 mins? (actually the second phase might last longer)
There are no control animals at all (not receiving formalin). It is possible that the behavioral effect of the compound is not related to its potential anti inflammatory propriety (eg. sedating effect). To rule that out total ambulatory distance should be also measured in control and formalin treated animals with and without compound pretreatment.
At least some in vivo pharmacokinetic data should be given
Minor issues:
On figures 4 and 5 cAMP is AMPc
English should be rechecked.
Author Response
REVIEWER 1:
Comments and Suggestions for Authors
The paper describes molecular hybrids of dypirone and NSAID as potential pharmaceutical candidates for pain management. The research is interesting. However, there are a series of methodological questions to be addressed by the authors.
Major issues:
2.3. Section of the paper should be rephrased to have more insight about the methodical procedure. I believe that the pharmaceutical compounds are given after forskolin treatment, but its not elaborated in the manuscript. Neither the number of measurements is given nor the SD/SEM and the results of the statistical analysis are indicated in Fig 4 and 5. The figures should be at least partially self-explanatory, but this is not the case here.
There is no detailed description of the results of LASSBio-2265 as a CB2 inverse agonist. (eg. how many measurements are done). The same problem is seen in the radioligand experiments as well.
AUTHOR’S ANSWER: Thanks for this observation. The text of Section 2.3 has been reformulated to provide a better understanding. Experiments were performed in a CRO (Eurofins CEREP, Celle-Lévescault, France) by independent experimenters blinded to compound chemical identity. Data represents cyclic AMP measurements after cell stimulation with 10 μM forskolin, as explained in the current version of the manuscript. Negative and blank controls were performed in triplicate with and without the addition of forskolin, respectively. Positive controls used were CB agonists CP-55,940 (at 1 nM) and WIN 55,212-2 (at 100 nM). Each substance from the LASSBio series was tested in two concentrations (1 and 10 μM) as a single replicate, as presented in Figures 4 and 5. Such measurements were performed in order to confirm the planned activities, and effects greater than 25% from controls were indicative of agonist activity, as suggested by the experience of the CRO team. The raw data of these experiments were included in the Supplementary material. Similarly, texts of radioligand binding and inverse agonist assays were expanded to provide better understanding.
2.4. Please describe the methodology in a more detailed way. Were the behavior measurements in the 2nd phase were done between 15-30 min? (actually the second phase might last longer)
AUTHOR’S ANSWER: The methodology was rewritten in order to furnish more details about the experiment. Although the second phase of the formalin test can last longer than 30 minutes, the behavioral responses found after this time usually do not vary much according to the specialized literature [1,2]. Besides, measurements carried out between 15 and 30 minutes allow direct comparison to other cannabinoidergic modulators evaluated in the literature [3,4].
[1] Dubuisson D, Dennis SG. The formalin test: a quantitative study of the analgesic effects of morphine, meperidine, and brain stem stimulation in rats and cats. Pain. 1977 Dec;4(2):161-174.
[2] Tjølsen A, Berge OG, Hunskaar S, Rosland JH, Hole K. The formalin test: an evaluation of the method. Pain. 1992 Oct;51(1):5-17.
[3] Burgos E, Pascual D, Martín MI, Goicoechea C. Antinociceptive effect of the cannabinoid agonist, WIN 55,212-2, in the orofacial and temporomandibular formalin tests. Eur J Pain. 2010 Jan;14(1):40-8.
[4] Cascio MG, Bolognini D, Pertwee RG, Palazzo E, Corelli F, Pasquini S, Di Marzo V, Maione S. In vitro and in vivo pharmacological characterization of two novel selective cannabinoid CB(2) receptor inverse agonists. Pharmacol Res. 2010 Apr;61(4):349-54.
- There are no control animals at all (not receiving formalin). It is possible that the behavioral effect of the compound is not related to its potential anti-inflammatory propriety (eg. sedating effect). To rule that out total ambulatory distance should be also measured in control and formalin-treated animals with and without compound pretreatment.
AUTHOR’S ANSWER: We thank you for the suggestion. However, it must be emphasized that no visible effect on ambulatory activity was observed during experimentation. Besides, substance administration occurred 60 minutes before the injection of formalin, which is a reasonable time for throughout absorption and distribution of lipophilic substances to the central nervous system (as occurs with morphine). Therefore, if any sedation was to be observed, a reduction in mice reactivity in the first phase of the test would also be expected. As this was not the case, any significant sedative effect was ruled out.
-At least some in vivo pharmacokinetic data should be given
AUTHOR’S ANSWER: We thank you for the suggestion. However, this work accounts for the first proof-of-concept assays for the LASSBio series of pyrazolamides. Physicochemical and druggability properties were performed for the prediction of compound pharmacokinetics (GI absorption and distribution to the CNS) in order to guide further optimizations in the future. In vivo pharmacokinetic studies are warranted.
Minor issues:
- On figures 4 and 5 cAMP is AMPc
AUTHOR’S ANSWER: Thanks, we corrected this notation.
-English should be rechecked.
AUTHOR’S ANSWER: The manuscript has been revised for better understanding and It was checked for a native English speaker from American Journal Experts.
Reviewer 2 Report
Article on “2-Arylpropionic Acid Pyrazolamides as Cannabinoid CB2 2 Receptor Inverse Agonists Endowed with Anti-inflammatory 3 Properties” by Daniela R. de Oliveira et al was found to interesting research work, may consider for acceptance after minor revision.
Comment 1: Authors have synthesised compounds (7-11). However, details of separation (chromatographic techniques) for these compounds are missing.
Comment 2: Correct the statement “However, the affinity obtained for compound (11) is on the same order of magnitude as those found by Rogosh and coworkers for arachidonyl-pyrazolamides (5) and (6), as illustrated in Table 2”.
Comment 3: Compounds (7-11) have studied for permeability by PAMPA-BBB model. Due to presence of hydrophobic groups, these compounds may have bioavailability issue. Please mention optimum concentration for cell bioavailability.
Comment 4: Compounds (7-11) may be studied for competitive enzyme (responsible for the cause inflammation) inhibition for further increasing the quality of the paper. You may refer below articles for the details. If possible, include other pathways that are targeted by these compounds.
Li F, Shanmugam MK, Chen L, Chatterjee S, Basha J, Kumar AP, Kundu TK, Sethi G. Garcinol, a polyisoprenylated benzophenone modulates multiple proinflammatory signaling cascades leading to the suppression of growth and survival of head and neck carcinoma. Cancer Prev Res (Phila). 2013 Aug;6(8):843-54. doi: 10.1158/1940-6207.CAPR-13-0070.
Abd El-Karim SS, Mohamed HS, Abdelhameed MF, El-Galil E Amr A, Almehizia AA, Nossier ES. Design, synthesis and molecular docking of new pyrazole-thiazolidinones as potent anti-inflammatory and analgesic agents with TNF-α inhibitory activity. Bioorg Chem. 2021 Jun;111:104827. doi: 10.1016/j.bioorg.2021.104827.
Ju Z, Li M, Xu J, Howell DC, Li Z, Chen FE. Recent development on COX-2 inhibitors as promising anti-inflammatory agents: The past 10 years. Acta Pharm Sin B. 2022 Jun;12(6):2790-2807. doi: 10.1016/j.apsb.2022.01.002.
Comment 4: Synthesised compounds was confirmed by spectral data. Cross check with literature on 13C-NMR spectral data of fluorinated compounds (For example: Lasso Bio-2262). Mention the 13C-spectral data accordingly.
Comment 5: Please mention the optimum concentration for toxicity of compounds.
Comment 6: Conclusion part can be modified for better understanding.
Comment 7: Title may modify as “Synthesis of 2-Arylpropionic Acid Pyrazolamides as Cannabinoid CB2 Receptor Inverse Agonists Endowed with Anti-inflammatory Properties.
Comment 7: Clarity of the spectral images can be increased.
Comment 8: Author must correct the typographic mistakes of the manuscript before submitting revised manuscript.
Comment 9: Check once more for the stability of the compounds (7-11), as these compounds are having sterically hindered groups on terminal Nitrogen atom.

Author Response
REVIEWER 2:
Comments and Suggestions for Authors
Article on “2-Arylpropionic Acid Pyrazolamides as Cannabinoid CB2 2 Receptor Inverse Agonists Endowed with Anti-inflammatory 3 Properties” by Daniela R. de Oliveira et al was found to interesting research work, may consider for acceptance after minor revision.
Comment 1: Authors have synthesised compounds (7-11). However, details of separation (chromatographic techniques) for these compounds are missing.
AUTHOR’S ANSWER: We further described the purification of the obtained products for clarifying the methods used. Used chromatographic procedures were already established and standardized in our laboratory. Briefly, compounds were purified by isocratic elution with ethyl acetate:hexane (9:1, v/v) from a flash silica gel column (1077341003; Millipore, USA), using aprox. 1 gram of silica per mg of raw product.
Comment 2: Correct the statement “However, the affinity obtained for compound (11) is on the same order of magnitude as those found by Rogosh and coworkers for arachidonyl-pyrazolamides (5) and (6), as illustrated in Table 2”.
AUTHOR’S ANSWER: The sentence was rewritten to: “However, the affinity obtained for compound (11) is on the same magnitude as those found by Rogosh and coworkers for arachidonyl-pyrazolamides (5) and (6), as illustrated in Table 2”
Comment 3: Compounds (7-11) have been studied for permeability by the PAMPA-BBB model. Due to the presence of hydrophobic groups, these compounds may have bioavailability issues. Please mention the optimum concentration for cell bioavailability.
AUTHOR’S ANSWER: The aqueous solubility of pyrazolamides was determined in buffer solution at pH 7.4, yielding values within​​ the range of 10 to 100 µM, which is considered a reasonable range since aqueous solubility values ​​greater than 100 µM can result in permeability limitations and consequent reductions in intestinal lumen absorption [1,2]. The lipophilicity values ​​(log D7.4) of pyrazolamides (7-11) were determined by the shake-flask method using phosphate buffer at pH 7.4 [3] and the values ​​obtained are within the range considered ideal for a drug candidate, i.e., 2-3 [4]. Solubility and lipophilicity at ideal values ​​constitute desired pharmacokinetic parameters for the development of drug candidates for oral administration, being indications of good bioavailability.
[1] Cisneros, J.A.; Robertson, M.J.; Mercado, B.Q.; Jorgensen, W.L. Systematic Study of Effects of Structural Modifications on the Aqueous Solubility of Drug-like Molecules. ACS Med. Chem. Lett. 2017, 8, 124–127, doi:10.1021/acsmedchemlett.6b00451.
[2] Faller, Bernard, Desrayaud, Sandrine, Berghausen, Joerg, Laisney, Marc and Dodd, Stephanie. "4. How solubility influences bioavailability". Solubility in Pharmaceutical Chemistry, edited by Christoph Saal and Anita Nair, Berlin, Boston: De Gruyter, 2020, pp. 113-132. https://doi.org/10.1515/9783110559835-004
[3] Sharapova, A.; Ol’khovich, M.; Blokhina, S.; Perlovich, G.L. Experimental Examination of Solubility and Lipophilicity as Pharmaceutically Relevant Points of Novel Bioactive Hybrid Compounds. Molecules 2022, 27, 6504. https://doi.org/10.3390/molecules27196504
[4] Arnott JA, Planey SL. The influence of lipophilicity in drug discovery and design. Expert Opin Drug Discov. 2012 Oct;7(10):863-75. doi: 10.1517/17460441.2012.714363.
Comment 4: Compounds (7-11) may be studied for competitive enzyme (responsible for the cause inflammation) inhibition for further increasing the quality of the paper. You may refer below articles for the details. If possible, include other pathways that are targeted by these compounds.
AUTHOR’S ANSWER: Thanks in advance for your suggestion. The in vitro experiments performed for the characterization of the inverse agonist profile at CB2 receptors were performed as previously described in the literature [1,2]. In addition, CB2 receptors are a GPCR Gi/Go family that inhibits adenylyl cyclase and decreases cAMP synthesis. In transfected CHO cells, the CB2 receptor does not couple to either phospholipases (A2, C or D) or Ca2+ mobilization [3]. Moreover, cannabinoid agonists inhibit cAMP accumulation in these cells, while no effect is reported in untransfected cells [3]. For in vivo studies, analgesic effects are already reported for CB2 inverse agonists, which do not induce hyperalgesia but, in fact, decrease pain behavior in the formalin test in rodent models [2,4-8].
[1] Ross et al. Br J Pharmacol, 126: 665-672, 1999. DOI: 10.1038/sj.bjp.0702351
[2] Hanus et al. Proc Natl Acad Sci U S A, 96(25): 14228-33, 1999. DOI: 10.1073/pnas.96.25.14228
[3] Felter et al. Mol Pharmacol, 48 (3): 443-50, 1995.
[4] Finn et al. Pain, 162: S5-S25, 2021. DOI: 10.1097/j.pain.0000000000002268
[5] Beaulieu et al. Eur J Pharmacol., 396(2-3): 85-92, 2000. DOI: 10.1016/s0014-2999(00)00226-0
[6] Cascio et al. Pharmacol Res, 61(4): 349-54, 2009. DOI: 10.1016/j.phrs.2009.11.
[7] Pasquini et al., J Med Chem 2012 55 (11), 5391-5402 DOI: 10.1021/jm3003334
Comment 4: Synthesised compounds was confirmed by spectral data. Cross check with literature on 13C-NMR spectral data of fluorinated compounds (For example: Lasso Bio-2262). Mention the 13C-spectral data accordingly.
AUTHOR’S ANSWER: Thank you for the suggestion, the 13C-NMR assignment was included throughout the text in section 3.2.
Comment 5: Please mention the optimum concentration for toxicity of compounds.
AUTHOR’S ANSWER: In the in vivo assay, no signs of sedation or behavioral changes were observed during the formalin assay. Therefore, no explicit signs of toxicity have been detected at this time. As these assays are proof of concept, the toxicity of these compounds will be evaluated in future studies.
Comment 6: The conclusion part can be modified for better understanding.
AUTHOR’S ANSWER: The conclusion was rewritten for better understanding.
Comment 7: Title may modify as “Synthesis of 2-Arylpropionic Acid Pyrazolamides as Cannabinoid CB2 Receptor Inverse Agonists Endowed with Anti-inflammatory Properties.
AUTHOR’S ANSWER: We appreciate the suggestion to change the title of the work but as this work also focuses on aspects of cannabinoid pharmacology, we decided to maintain the submitted title.
Comment 7: Clarity of the spectral images can be increased.
AUTHOR’S ANSWER: Thanks a lot for observation. Spectral images were resubmitted in higher resolution.
Comment 8: Author must correct the typographic mistakes of the manuscript before submitting revised manuscript.
AUTHOR’S ANSWER: We are grateful for the observations and have reviewed the manuscript prior to submission.
Comment 9: Check once more for the stability of the compounds (7-11), as these compounds are having sterically hindered groups on terminal Nitrogen atom.
AUTHOR’S ANSWER: Chemical stability was performed according to standardized protocols based on OECD and FDA recommendations and no signs of chemical degradation were observed.
Reviewer 3 Report
The article “2-Arylpropionic Acid Pyrazolamides as Cannabinoid CB2 Receptor Inverse Agonists Endowed with Anti-inflammatory Properties” authored by Daniela R. de Oliveira et al. meets the main criteria of Pharmaceuticals and could be published in this journal after some major points will be addressed:
1. The authors should explain why did they use so unusual ratio of reagents (1 eq. of carboxylic acid, 1.2 eq. of 1,1'-carbonyldiimidazole and hydroxybenzotriazole, 2 eq. of triethylamine and 1.5 eq. of 4-amino-antipyrine) for synthesis of compounds 7-9? Actually, the yields of target compounds are moderate but I believe they can be improved by changing the reaction conditions including the reagents ratio.
2. Numeration of atoms in target molecules should be provided, at least in the Supporting information. It’s hardly possible to perceive NMR data without it.
3. The NMR spectra description must be improved. For example, on p. 11, line 307 the authors provide the following signal assignment: “7.51-7.08 (m, 9H, H4, H5, H5', H6', H7')”. However, from the corresponding spectrum in the SI it can be seen that there are at least three distinguishable multiplets in this region.
Almost the same with 13C NMR spectra. Such kind assignment: “152.01-114.79 (C4, C5, C8, C9, C10, C5', C6', C7')” is not appropriate. All carbon signals should be provided and separated by commas whether or not you can assign them. However, in such small organic molecules most of proton and carbon signals can be easily assigned using 2D NMR techniques.
Also, it’s not clear why the authors in some cases use this kind of description: “0.86 and 0.84 (s, 3H, H9; s, 3H, H10)”. There are two singlet signals which should be assigned separately.
The aforementioned mistakes can be found in almost all 1H and 13C spectra descriptions and should be fixed.
4. 1H and 13C NMR spectra of compounds 10-11 (especially, 11) demonstrate the presence of some side signals, which were not processed by the authors but can be seen with a naked eye. This fact raises a question about the purity of the products. The authors should carefully explain the reason for the appearance of these signals in the spectra.
5. In all HRMS descriptions the authors should provide the calculated mass of molecular ion along with the found value.
6. The performance of the SI should be improved. There are a lot of typing, grammatical and formatting errors.
Author Response
REVIEWER 3:
Comments and Suggestions for Authors
The article “2-Arylpropionic Acid Pyrazolamides as Cannabinoid CB2 Receptor Inverse Agonists Endowed with Anti-inflammatory Properties” authored by Daniela R. de Oliveira et al. meets the main criteria of Pharmaceuticals and could be published in this journal after some major points will be addressed:
- The authors should explain why did they use so unusual ratio of reagents (1 eq. of carboxylic acid, 1.2 eq. of 1,1'-carbonyldiimidazole and hydroxybenzotriazole, 2 eq. of triethylamine and 1.5 eq. of 4-amino-antipyrine) for the synthesis of compounds 7-9? Actually, the yields of target compounds are moderate but I believe they can be improved by changing the reaction conditions including the ratio of the reagents.
AUTHOR’S ANSWER: The conditions of the reactions carried out in this work were empirically optimized in order to obtain the product of interest in good yields from adjustments based on literature data.
[1] Ghosh AK, Shahabi D. Synthesis of amide derivatives for electron-deficient amines and functionalized carboxylic acids using EDC and DMAP and a catalytic amount of HOBt as the coupling reagents. Tetrahedron Lett. 2021 Jan 19;63:152719.
[2] Valeur E, Bradley M. Amide bond formation: beyond the myth of coupling reagents. Chem Soc Rev. 2009 Feb;38(2):606-31.
[3] Joshua R. Dunetz, Javier Magano, and Gerald A. Weisenburger Large-Scale Applications of Amide Coupling Reagents for the Synthesis of Pharmaceuticals Org. Proc. R & D 2016 20 (2), 140-177
- Numeration of atoms in target molecules should be provided, at least in the Supporting information. It’s hardly possible to perceive NMR data without it.
AUTHOR’S ANSWER: Thanks and sorry for the error in the submitted version, we corrected it and included NMR spectra better elucidated.
- The NMR spectra description must be improved. For example, on p. 11, line 307 the authors provide the following signal assignment: “7.51-7.08 (m, 9H, H4, H5, H5', H6', H7')”. However, from the corresponding spectrum in the SI it can be seen that there are at least three distinguishable multiplets in this region.
Almost the same with 13C NMR spectra. Such kind assignment: “152.01-114.79 (C4, C5, C8, C9, C10, C5', C6', C7')” is not appropriate. All carbon signals should be provided and separated by commas whether or not you can assign them. However, in such small organic molecules, most proton and carbon signals can be easily assigned using 2D NMR techniques.
Also, it’s not clear why the authors in some cases use this kind of description: “0.86 and 0.84 (s, 3H, H9; s, 3H, H10)”. There are two singlet signals which should be assigned separately.
The aforementioned mistakes can be found in almost all 1H and 13C spectra descriptions and should be fixed.
AUTHOR’S ANSWER: The authors thank you for your observations. All spectra were revised and assignments and markings were corrected according to the standards of the literature.
- 1H and 13C NMR spectra of compounds 10-11 (especially, 11) demonstrate the presence of some side signals, which were not processed by the authors but can be seen with the naked eye. This fact raises a question about the purity of the products. The authors should carefully explain the reason for the appearance of these signals in the spectra.
AUTHOR’S ANSWER: Thanks for the observation and we apologize for the confusion, the fact is that the decoplanarization of the conjugated system of the compounds leads to more than one established conformation in NMR spectra, as demonstrated in by nuclear magnetic resonance. Other characterization techniques were performed in order to certify compound purity, like melting point determination by differential scanning calorimetry, relative purity by high-performance liquid chromatography, Fourier transform infrared spectra (obtained by attenuated total reflectance) for functional group characterization, and high-resolution mass spectrum, which further corroborated compound purity by the experimental mass of molecular ion peaks. In this revised version we further elaborated on 1H-NMR spectrum elucidation for better understanding.
- In all HRMS descriptions, the authors should provide the calculated mass of molecular ions along with the found value.
AUTHOR’S ANSWER: Thanks, we provided the calculated mass of molecular ions along with the experimental value in the revised version (section 3.2) and in supplementary data. Compound (7): [M+H]+ (calculated): 392.23325 m/z, [M+H]+ (experimental): 392.23268 m/z; Compound (8): [M+H]+ (calculated): 430.19253 m/z, [M+H]+ (experimental): 430.19174 m/z; Compound (9): [M+H]+ (calculated): 398.18630 m/z, [M+H]+ (experimental): 398.18582 m/z; Compound (10): [M+H]+ (calculated): 406.24890 m/z, [M+H]+ (experimental): 406.24832 m/z and Compound (11): [M+H]+ (calculated): 444.20818 m/z, [M+H]+ (experimental): 444.20806 m/z.
- The performance of the SI should be improved. There are a lot of typing, grammatical and formatting errors.
AUTHOR’S ANSWER: Thanks for your careful observations, we made corrections for better understanding.
Reviewer 4 Report
I suggest the authors to evaluate if the newly synthetized compounds tested in vivo have activity on COX enzymes and on FAAH. This is necessary to confirm the proposed mechanism of action (CB2 inverse agonism) accounting for the observed antiinflammatory activity (at 3 mg/kg). This can be done as a service without specific difficulties.
Author Response
REVIEWER 4:
Comments and Suggestions for Authors
I suggest the authors to evaluate if the newly synthetized compounds tested in vivo have activity on COX enzymes and on FAAH. This is necessary to confirm the proposed mechanism of action (CB2 inverse agonism) accounting for the observed antiinflammatory activity (at 3 mg/kg). This can be done as a service without specific difficulties.
AUTHOR’S ANSWER: We appreciate the suggestion regarding further testing. Evidence for inverse agonism of cannabinoid receptors is stronger than direct action on arachidonic acid metabolism. First, the analgesic effects of dipyrone in vivo are blocked by cannabinoid antagonists [1-3] and inverse agonism of CB2 receptors does not affect dipyrone-induced analgesia in mice [4] and even produce analgesia to algic stimuli in rats [5,6]. Moreover, pyrazolamides are already reported to be cannabinoid ligands with low selectivity between CB1 and CB2 isoforms [7]. LASSBio-2265 affinity for CB2 receptors is similar to previously reported CB1/CB2 affinities for metabolites 5 and 6 [7]. Moreover, the binding assay performed at CB2 receptors subsides the inverse agonist hypothesis for the increased cAMP accumulation seen in CHO-CB2 cells with these compounds.
[1] Santos et al. Eur J Pharmacol., 874: 173005, 2020. DOI: 10.1016/j.ejphar.2020.173005
[2] Crunfli, Vilela & Giusti-Paiva. Clin Exp Pharmacol Physiol, 42(3): 246-55, 2015. DOI:10.1111/1440-1681.12347
[3] Elmas & Ulugol. J Neural Transm (Vienna), 120(11): 1533-8, 2013. DOI: 10.1007/s00702-013-1052-7
[4] Schlosburg et al. Behav Pharmacol., 23(7): 722-6, 2012. DOI: 10.1097/FBP.0b013e3283584794
[5] Finn et al. Pain, 162: S5-S25, 2021. DOI: 10.1097/j.pain.0000000000002268
[6] Beaulieu et al. Eur J Pharmacol., 396(2-3): 85-92, 2000. DOI: 10.1016/s0014-2999(00)00226-0
[7] Rogosch et al. Bioorg Med Chem., 20(1): 101-107, 2012. DOI: 10.1016/j.bmc.2011.11.028
Round 2
Reviewer 1 Report
Although the manuscript has been improved the following (major) problems persist:
- a single measurement regarding effects of the pharmacological compounds is not enough for scientific evaluation
- Behavior studies need a naive control animal group (observations provided by the authors are not scientific results)
Reviewer 4 Report
I'm satisfied with this new version